# Ferrocifen Loaded Lipid Nanocapsules: A Promising Anticancer Medication against Multidrug Resistant Tumors

**DOI:** 10.3390/cancers13102291

**Published:** 2021-05-11

**Authors:** Pierre Idlas, Elise Lepeltier, Gérard Jaouen, Catherine Passirani

**Affiliations:** 1Micro & Nanomedecines Translationnelles (MINT), University of Angers, Inserm, The National Center for Scientific Research (CNRS), SFR ICAT, 49000 Angers, France; pierre.idlas@univ-angers.fr (P.I.); elise.lepeltier@univ-angers.fr (E.L.); 2Sorbonne Universités, Université IPCM, Paris 6, UMR 8232, IPCM, 4 place Jussieu, 75005 Paris, France; gerard.jaouen@chimieparistech.psl.eu; 3PSL University, Chimie ParisTech, CNRS, Institut de Recherche de Chimie Paris, 75005 Paris, France

**Keywords:** nanomedicine, bioorganometallic drug, cancer, preclinical trial, combination therapy

## Abstract

**Simple Summary:**

One of the primary causes of chemotherapy failure is the occurrence of cancer multidrug resistance (MDR). Uncontrolled growth of resistant tumor cells leads to metastasis and recurrence, associated with high mortalities. Ferrocifens have been shown to possess anticancer properties via an original mechanism dependent on redox properties and generation of active metabolites that can cause disruption of cell metabolism. However, these molecules are highly insoluble in water, requiring a formulation stage before being administered in vivo. Lipid nanocapsules (LNCs) have already demonstrated their ability to successfully encapsulate various hydrophobic therapeutic agents, such as ferrocifens, and offer the option of surface modification, making it possible to adapt the pharmacological behavior of the nanocarrier. The aim of this review is to give, for the first time, an overview of the in vitro and in vivo studies performed with ferrocifen-loaded LNCs on several MDR cancers.

**Abstract:**

Resistance of cancer cells to current chemotherapeutic drugs has obliged the scientific community to seek innovative compounds. Ferrocifens, lipophilic organometallic compounds composed of a tamoxifen scaffold covalently bound to a ferrocene moiety, have shown very interesting antiproliferative, cytotoxic and immunologic effects. The formation of ferrocenyl quinone methide plays a crucial role in the multifaceted activity of ferrocifens. Lipid nanocapsules (LNCs), meanwhile, are nanoparticles obtained by a free organic solvent process. LNCs consist of an oily core surrounded by amphiphilic surfactants and are perfectly adapted to encapsulate these hydrophobic compounds. The different in vitro and in vivo experiments performed with this ferrocifen-loaded nanocarrier have revealed promising results in several multidrug-resistant cancer cell lines such as glioblastoma, breast cancer and metastatic melanoma, alone or in combination with other therapies. This review provides an exhaustive summary of the use of ferrocifen-loaded LNCs as a promising nanomedicine, outlining the ferrocifen mechanisms of action on cancer cells, the nanocarrier formulation process and the in vivo results obtained over the last two decades.

## 1. Introduction

Nanomedicine is a multidisciplinary field embracing researchers from chemistry, galenic pharmacy, biochemistry and biology, with the aim of developing innovative therapies using biologically active molecules (drugs) formulated in nanoparticles (NPs) to treat a panel of diseases such as cancer, inflammatory diseases or neurological disorders. The field of oncology is predominant and has been of increasing interest since the anticancer nanomedicine Doxil^®^ became the first to win FDA approval [1,2,3]. Over recent decades, many small active molecules (cisplatin, doxorubicin, tamoxifen, paclitaxel, etc.) were synthesized and are now used in current chemotherapy treatments. For example, platinum-based chemotherapy, akin to cisplatin, has been used alone or in combination in 70% of cancer treatments [4]. The main target of platinum coordination complexes and doxorubicin is DNA, leading to the apoptosis of cells. However, one drawback of these current treatments is the resistance developed by some cancer cells to proapoptotic stimuli, for example, in glioblastoma and melanoma, resulting in the ineffectiveness of these molecules [4,5]. Tamoxifen, the usual treatment molecule for breast cancer, is known as a selective estrogen receptor modulator (SERM). Conversely, its nonselectivity against hormone-independent cancer such as triple-negative breast cancer significantly reduces the spectrum of its applications [6]. Paclitaxel is a microtubule-stabilizing drug able to block cell cycle progression, thus inhibiting the growth of cancer cells, with interesting recent results observed in immunotherapy [7]. However, many studies have proved that paclitaxel is a substrate of the membrane-bound drug efflux pump P-glycoprotein (P-gp) and shows limited efficacy against the resistant tumors [8].

In the late 1980s, Jaouen et al. synthesized a new class of bio-organometallic molecules [9,10]. The idea was to use an organometallic complex based on iron, ruthenium or rhodium covalently bound to a tamoxifen scaffold, which could offer new curative properties with a wide range of biological applications due to its direct metal-carbon covalent bond. The potential efficacy of these innovative compounds is conferred by the chemical reactivity of the organometallic complex, different from that of the metal or the organic ligands alone. Amongst these complexes, the ferrocifen family consists of a ferrocene moiety covalently bound to a tamoxifen skeleton that has shown promising in vitro results against MDR cancer cell lines [11,12,13]. However, ferrocifen compounds are hydrophobic and need a suitable formulation strategy for intravenous injection, the most commonly used route of administration in cancer patients.

Three generations of nanoparticles (NPs) can be differentiated. In the 1970s, the first nanomedicines were biodegradable but accumulated mainly in the liver. The second generation is characterized by the surface modification with hydrophilic molecules, mainly polyethylene glycol (PEG), in order to confer stealth properties and to passively accumulate in solid tumor tissues via the so-called enhanced permeation and retention (EPR) effect [14,15]. However, the EPR effect still divides the scientific community [16,17]. Explored since the 1990s [18,19,20], the third generation of nanomedicines concerns their surface modification by an active ligand such as a monoclonal antibody or a peptide for specific cancer cell targeting called “active targeting”. Since the beginning of the 21st century, these innovative NPs have received the most attention, with this surface modification aiming to improve the biodistribution and limit the side effects of the drugs [21,22,23]. Within this array of nanocarriers, Heurtault et al. patented a hybrid structure between polymer NPs and liposomes, called lipid nanocapsules (LNCs). LNCs are prepared by a solvent-free process and are composed of an oily core surrounded by a shell of lecithin and PEGylated surfactants [24,25]. More than a hundred studies have confirmed the real promise of this NPs as a drug delivery system, especially suitable for the ferrocifen molecules [26,27,28,29,30,31,32].

The aim of this review is to give, for the first time, an exhaustive overview on the in vitro and in vivo studies performed with ferrocifen-loaded LNCs. Firstly, ferrocifen mechanisms of action at the cellular level will be detailed in relation to potential anti-MDR activity. Then, the specificities of LNCs will be described, especially through their potential as ferrocifen nanocarriers, and the in vivo experiments on MDR cancers such as glioblastoma, metastatic melanoma and breast cancer will be reviewed.

## 2. Ferrocifens: Promising Molecules against Cancer

Ferrocifen compounds were synthesized for the first time in 1996 by Jaouen et al. [33,34]. The general molecular structure is composed of a tamoxifen scaffold bound to a ferrocene moiety (Figure 1). The initial idea was to use an old drug, tamoxifen, and add a metallocene such as ferrocene or ruthenocene in order to confer new properties on the molecule. The ferrocifen family exhibited the best bioactivity [35].

The ease of ferrocene functionalization and the ability to shape the overall structure by adding a basic side chain (Figure 1, group R_1_) or to change the functional group R_2_ has allowed the synthesis of more than three hundred complexes over the last two decades, some of them showing anticancer properties on a large panel of MDR cancer cell lines including glioblastoma, metastatic melanoma, breast cancer and leukemia [36,37,38,39]. Indeed, structure–activity relationship studies have allowed for the identification of the best potential candidates for in vivo studies. In this section, the link between the structure of 5 ferrocifen molecules (P5, P15, P53, P722 and DP1, see Table 1) and their biological efficiency on cancer cells will be detailed.

### 2.1. Ferrocifen Mechanisms of Action

Since the first synthesis of ferrocifen compounds was performed, the mechanism of action has been increasingly well-understood. It has been shown to vary depending on whether the molecule is used as an anticancer agent or as immunomodulator or antiparasitic compounds [40,41]. This review focuses on the anticancer properties of ferrocifens. All the studies performed have demonstrated that the cell death pathway induced by these organometallic molecules differs according to the structure of the complexes, the cell lines tested and the ferrocifen concentration [41]. The use of these complexes is particularly potent as they can confer both antiproliferative and cytotoxic effects, as schematically summarized in Figure 2. The formation of the quinone methide (QM) and its effects at the cellular level, discussed subsequently, are elements needed to be determined to understand this phenomenon.

#### 2.1.1. Formation of the Quinone Methide

The formation of quinone methide (QM) can only occur if the ferrocenyl group is located on carbon 2 of the but-1-ene group and if the phenol is on carbon 1 (Figure 1). This configuration allows a conjugated π-system between the ferrocenyl and the phenol groups [11,42]. Moreover, the unique redox properties of ferrocene could lead to the reversible oxidation of Fe(II) to Fe(III) in the presence of Reactive Oxygen Species (ROS), molecules overexpressed in cancer cells. Indeed, the ferrocene has a redox potential similar to the redox potential found in the cell and is thus favorable for in vitro electron transfer [43]. Then, as shown in Figure 3, a redox process involving two electrons and two protons leads to the formation of a phenoxy radical and then to the QM. All the steps presented in Figure 3 have been confirmed by electrochemistry and electron paramagnetic resonance [42,44]. The authors of these studies attribute this QM formation to the role of the ferrocene as a “redox antenna”, followed by an electronic delocalization [45,46]. However, according to the initial ferrocifen structure, the QMs obtained for P5, P15, P53 and P722 will not have the same structures, as presented in Table 1, and this difference will confer variable biological efficacy [47,48,49]. Indeed, P53 showed a better efficacy than P5 and P15 on several MDR cancer cell lines such as MDA-MB-231 triple-negative breast cancer and cisplatin-resistant A2780cisR human ovarian cancer cells, due to its ability to form two QMs: a vinyl QM such as P5 or P15 but also a QM involving a furan heterocycle, conferring good stability on its structure. From these two QMs, several metabolites could be produced in vitro and bind to various cell targets, such as glutathione [50]. For P722, its exceptional antiproliferative effect may be explained by the atypical lone pair-π interaction between one carbonyl group of the imide and the center of the aromatic QM ring, responsible for an improvement of the stability of the QM-P722 [48]. No QM could be isolated from DP1 (ansa-FcdiOH) by electron paramagnetic resonance spectroscopy studies; more precisely, its QM was too unstable, with a short half-life. Nevertheless, the high cytotoxicity of DP1 compared to the acyclic series P5 and P15 could come from the formed free radical QM [51,52].

#### 2.1.2. Ferrocifen Mechanisms of Action at the Cellular Level

Early studies proposed some possible tumor cell mechanisms due to the well-known antiestrogenic effect of ferrocifens (cytostatic effect) [12,53]. Interestingly, ferrocifens can also drive senescence (another cytostatic effect) as well as apoptosis (cytotoxic effect), depending on the initial concentration used [41,54]. Senescence is an irreversible arrest of proliferation via different pathways, such as genomic damage caused by the activation of p53, a tumor-suppressor protein controlling the expression of numerous activation factors in cells. Table 2 and Table 3 show the values of half maximal inhibitory concentration IC_50_ (concentration inhibiting 50% of a specific activity of a biological target) and of growth inhibition concentration GI_50_ (concentration inhibiting 50% of cell proliferation), found for the different ferrocifens: all the different values are in the low or submicromolar range. This can be explained by the multiple pathways of activity, as discussed below.

##### Antiestrogenic Proliferative Effect

The antiestrogenic proliferative effect of the ferrocifens has been well-discussed in the literature. With their molecular structures, they can bind to the estrogen receptor, filling the hydrophobic pocket [60] and, by modulating the expression of oxidative stress proteins, block the binding of 17β-oestradiol, which plays an important role in regulating hydrogen peroxide levels in ERα+ cells [61]. This can lead to an interruption of breast cancer cell proliferation. Even if the relative binding affinity (RBA) of ferrocifens for the estrogen receptor is lower than with tamoxifen, probably due to the steric effect of the ferrocenyl group, the values are still satisfactory, at around 10% [38,41]. According to Paterni et al., a good ER-ligand should possess two OH groups (one should be a phenol or a phenol bioisostere) linked by a lipophilic central scaffold and placed at a distance of about 11 Å [62]; this is the case for P15, whose structure corresponds to this description and presents the highest RBA values (11.5%) compared to P5 (9.6%) [41]. Moreover, Jaouen et al. showed that modifying one phenol by adding a basic chain, such as -O(CH_2_)_3_NMe_2_ in P15 (group R_1_ in Figure 1), could improve the antiestrogen effect of ferrocifens [38,53,63]. Moreover, the length of this chain is an important parameter as no antiestrogenic effect was found with the basic chain -O(CH_2_)_2_N(CH_3_)_2_. This can be explained by the fact that the lone pair of the nitrogen needs to interact by hydrogen bonding with the Asp351 of the estrogen receptor to provide the right position in the ER pocket, leading to the correct ER conformation [38]. This is why P15, due to its structure, has a better antiestrogenic effect than P5. This antiestrogenic effect is only observed at low concentration (0.1 µM) and will induce senescence [50,52,54]; it is considered to be the first organometallic selective estrogen receptor modulator.

##### Cytotoxic Effect

The cytotoxic effect of the ferrocifens seems to be related to the unique redox properties of ferrocene, able to induce a reversible oxidation of Fe(II) to Fe(III) (Fenton reaction), followed by the production of ROS known for their bioreactivity [64,65], and related to the structure of the ferrocifen molecule [38,42,48,66,67]. Studies on compounds possessing other moieties instead of iron, such as ruthenium, titanium or rhenium, showed that these compounds did not produce the cytotoxic effect observed with ferrocifens at the same concentration (1 µM) [13].

As mentioned above, the formation of QM has been suggested as one of the principal causes of the cytotoxic effect [35,38,68]. It corresponds to the major active metabolite obtained in the reaction between the ferrocifen and the nicotinamide adenine dinucleotide phosphate (NADPH) present in the P450 cytochrome [68], except for DP1 (QM half-life too short). Depending on the initial ferrocifen used, other metabolites are obtained but their biological efficiency is less convincing or is still under investigation. It has been shown that the QM, being electrophilic, could react with various cell targets such as thiols from thioredoxin reductase (TrxR), via a Michael addition [36,52,56,67]. TrxR is a selenoenzyme permitting cellular redox regulation by activating transcription factors that alter gene expression and peroxiredoxins, allowing a decrease in ROS [61,69]. This enzyme is found both in cytosol and mitochondria of cells and is often overexpressed in cancer cells [61]. For DP1, as previously mentioned, electron paramagnetic resonance studies did not detect formation of the QM (or it was too unstable to be detected) but only the QM radical, which is also highly cytotoxic, especially on TrxR (IC_50_ = 0.15 µM) [52]. The higher cytotoxicity of DP1 compared to P5 or P15 on cancer cells could be explained as follows: DP1 can react with both the selenocysteine and thiols from TrxR, whereas P5 and P15 selectively react with only the selenol group [36,52]. Another example of how the ferrocifen structure can change the biological reactivity of the compound is presented by a comparison between P5 and P15. It has been shown that both could inhibit TrxR present in cytosol in MCF-7 and MDA-MB-231 breast cancer cells, but only P15 had showed inhibitive action on mitochondrial TrxR in Jurkat cells (leukaemia cell line) [36,67,70]. Indeed, whereas P5 induces a redox imbalance leading to lipid peroxidation in cytoplasm, P15 has an additional effect on mitochondria due to the dimethylamine propyl chain -O(CH_2_)_3_N(CH_3_)_2_, a lipophilic cation in the cancer cell environment, allowing a preferential localization in mitochondria. According to Scalcon et al., this overexpression of oxidized thioredoxin in mitochondria led to an increase in ROS and the release of cytochrome c, which can activate the caspase-cascade system, enzymes responsible for apoptosis [70,71,72].

Thus, ferrocifens could be seen as promising anticancer compounds as they provide a variety of mechanisms of action, depending on the cell line treated, the concentration used and their molecular structure. For example, by forming QM, ferrocifens behave as ROS-producing compounds by increasing the ROS level inside an oxidative stress environment. Moreover, it may be possible to consider ferrocifen as a prodrug: the corresponding QM inhibits cytosolic and mitochondrial TrxR more efficiently than the parent compound. However, in order to further analyze the efficacy of these molecules and to perform in vivo studies, ferrocifens need to be formulated in nanocarriers. Indeed, due to their high lipophilicity, they cannot be injected as a solution into the circulatory system and lipid nanocapsules could represent one potential answer to this.

## 3. Lipid Nanocapsule Characteristics

### 3.1. Composition and Formulation Process

Nanometric colloidal systems, as drug carriers, are dispersed systems, often oil in water dispersions, in which solid or liquid phases are in suspension in a fluid medium. The main process to obtain these systems is to use emulsification or nanoprecipitation methods. However, one drawback of these methods is the use of organic solvents, required to solubilize the excipients, mainly lipids or polymers [24]. Moreover, the size of nanoparticles (NPs) is one of the main parameters to be controlled for pharmaceutical applications, as they influence cell internalization and in vivo biodistribution. To avoid instability mechanisms, such as droplet coalescence, high amounts of surfactants and cosurfactants are often needed, leading to potential toxicity for human use. With the objective of optimizing these different parameters, Heurtault et al. developed a new process patented in 2001 [25]: lipid nanocapsules (LNCs) exhibiting a core–shell structure composed of a liquid oily core and an amorphous surfactant shell (Figure 4), without the use of any organic solvent. The formulation process is described in Figure 4a: a mixture of the various constituents is made, followed by three temperature cycles to pass through the phase inversion zone (PIZ) described below.

LNCs are composed of three main parts: an oily phase, an aqueous phase and nonionic surfactants. The oily phase is composed of caprylic/capric triglycerides (commercial name: Labrafac^®^ WR 1349), nonionic hydrophilic polyethylene glycol (15)-hydroxystearate surfactant (Solutol^®^ HS 15) and, as an amphiphilic cosurfactant, phosphatidylcholine from soybean (Lipoid^®^). The last principal component is the aqueous phase, MiliQ^®^ water, containing a fixed amount of sodium chloride (NaCl) to make the suspension injectable [24,25,26,27]. The size can be adjusted within a very narrow distribution. Thus, in order to obtain LNCs with an average size between 20 and 100 nm, the amounts of triglycerides, nonionic surfactants and water to be added are: 10–40% (*w*/*w*) of polyethylene glycol (15)-hydroxystearate; 35–80% (*w*/*w*) of water; 10–25% (*w*/*w*) of caprylic/capric triglycerides. It should be noted that each constituent has an influence on LNC formulation and stability [73]. For example, when the proportion of polyethylene glycol (15)-hydroxystearate increases, the average diameter and size distribution of the LNCs decrease. This phenomenon could be attributed to both the strong hydrophilic character of the nonionic surfactant due to the PEG part and the strong hydrophobic property from the stearate chain, bringing a great stability to the oil in water system (Figure 4b) [73]. According to its structure and properties, this molecule is responsible for the phase inversion of the system from oil in water to water in oil emulsion (Figure 4b). Briefly, the phase inversion temperature method is a low-energy emulsification method first introduced by Shinoda and Saito in 1969 [74]. This method is based on the specific ability of the usual nonionic surfactants, such as PEG surfactants, to modify their affinity for water and oil depending on temperature. As the temperature increases, the lipophilic property of nonionic surfactants becomes preponderant as a consequence of the dehydration of ethylene oxide groups from the PEG moiety. This dehydration is due to the rupture of the hydrogen bonds between oxide groups and water molecules, following the thermal agitation (Figure 4b). Thus, when the balance between lipophilicity and hydrophilicity appears in the phase inversion zone (PIZ), the system is a bicontinuous microemulsion and it is suddenly broken up by a rapid cooling step, immediately generating a nanoemulsion of small droplets constituting the LNCs (Figure 4a) [24,75]. Moreover, the use of polyethylene glycol (15)-hydroxystearate is of interest due to its ability to block P-glycoprotein (P-gp) related drug efflux, one of the main factors leading to tumor cell resistance in chemotherapy [76].

Concerning the surface structure of LNCs, several studies have shown that phosphatidylcholine is anchored in the oily phase, whereas the polyethylene glycol (15)-hydroxystearate is oriented towards the water phase (Figure 4c) [24,73,77].

### 3.2. Surface Modification (Passive and Active Targeting)

The principal drawback of NPs for drug delivery is the fact that these nanocarriers are recognized by the mononuclear phagocyte system (MPS) as foreign bodies. Indeed, after a systemic injection, the clearance of nanocarriers from the blood can be very rapid, due to the macrophages of the MPS, particularly Kupffer cells in the liver, and the complement system. Plasma protein such as C3b molecules (named opsonins), mostly present in blood, can bind to the surface of the NPs due to weak forces such as electrostatic and hydrophobic interactions, forming a protein corona (opsonization). Then, opsonins are recognized by macrophages, leading to the phagocytosis of the NPs [78,79,80,81]. However, one of the advantages of LNCs is their ability to carry out surface modifications, as shown in Figure 5, to improve the efficacy of the nanocarrier after administration. Several studies have shown the value of the PEG-based (DSPE-PEG_2000_) coating in the design of long-circulating LNCs [77,82,83,84,85,86]. PEG was chosen due to its physico-chemical characteristics: it is uncharged and hydrophilic [86,87]. Due to these characteristics, the PEG layer makes it possible to avoid the adsorption of opsonins and thus reduce clearance by the MPS [80,81]. Through surface modification of the LNCs with PEG, the nanocarriers acquire stealth properties, improving their blood circulation time and leading to a better accumulation in the tumor site. This is the so-called “passive targeting”. Indeed, Béduneau et al. have studied the stealth properties of LNCs decorated with PEG_1500_ stearate after an intravenous injection in a murine model. The results obtained revealed that this LNC modification allowed a very weak complement consumption and an increase in the blood circulation half-time, by reducing the protein adsorption, and hence the opsonization, due to an efficient steric hindrance [82]. These results were confirmed by other studies where PEG_2000_-coated LNCs improved the biodistribution in a murine model by extending the blood circulation time associated with good structural stability over several hours, after intravenous injection [26,83,88,89].

Moreover, the use of DSPE-PEG can be of interest for covalent functionalization, offering the possibility of shaping the terminal group of the molecule [90]. For example, Béduneau et al. used DSPE-PEG_2000_ functionalized with reactive-sulfhydryl maleimide groups (malDSPE-PEG) to perform an active targeting strategy via the covalent grafting of a homing moiety such as monoclonal antibody to the maleimide function [91]. In another study, Resnier et al. used the same molecule (malDSPE-PEG) to covalently bond a small artificial affinity protein named affitins, used as an alternative to antibodies [92]. Moreover, according to the structure of polyethylene glycol (15)-hydroxystearate possessing an acyl PEGylated group, a direct covalent bonding between this group and the amine on the N-terminal chain of a peptide can also be performed via a transacylation reaction. This process to prepare active functionalized LNCs by transacylation reaction was patented by Benoit and Perrier in 2010 [93]. Finally, the surface decoration can also be performed by a simple physico-chemical adsorption. This last strategy was particularly developed using a cell penetrating peptide, permitting improved capacity for NPs to cross cellular membranes with low toxicity and via energy-dependent and/or -independent mechanisms [28,94]. Indeed, Karim et al. used the NFL peptide, known to specifically enter glioblastoma cells and disturb the microtubule network [29], to functionalize the surface of ferrocifen-loaded LNCs [95]. Due to this functionalization, the uptake in U87MG cells (glioma cancer cell line) of the corresponding functionalized LNCs was higher than with conventional LNCs and lower in astrocytes (normal human astrocytes). This uptake seemed to be an energy-dependent process—a combination of micropinocytosis, clathrin-mediated and caveolin-mediated endocytosis [95].

## 4. Ferrocifen-Loaded LNCs: In Vitro Studies

Most of the ferrocifen compounds synthesized in recent decades have shown a promising efficacy on the National Cancer Institute (NCI) panel of 60 cancer cell lines (lung, breast, brain, ovary, kidney, prostate, melanoma, leukemia, colon, CNS, etc.), as presented in Table 2 and Table 3. 

Several studies demonstrated that the IC_50_ values were not affected by the encapsulation of ferrocifens into LNCs, whatever the formulation used. Indeed, Lainé et al. showed that the IC_50_ of free P15 on MDA-MB-231 breast cancer cells was 2 µM. The same value was found after LNC encapsulation [55]. For P53, Karim et al. obtained a similar IC_50_ on U87 MG glioma cells free or encapsulated into LNCs, with even a better efficacy after NFL peptide adsorption on P53 LNCs [95]. It was shown in this study that LNCs did not lead to any additional toxicity to the cancer cells. This was confirmed on SK-Mel28 melanoma cells by Resnier et al., where P5 and DP1 had the same toxicity profiles as free compounds or were encapsulated into LNCs, with no variation of the IC_50_ [59]. The same conclusion can be drawn for another melanoma cell line, B16F10, for free P722- or P722-loaded LNCs [96]. Moreover, several in vitro assessments showed that ferrocifens could induce both senescence and apoptosis depending on the concentration used. Indeed, Vessières et al. showed that P15 led to a higher percentage of senescence in both MCF-7 and MDA-MB-231 breast cancer cells by using a senescence-associated β-galactosidase assessment (one of the markers of senescence), compared to the tamoxifen (OH-TAM) [54], as shown in Figure 6a. The encapsulation of P15 in LNCs did not hamper its effect on MDA-MB-231 cells, as revealed by Lainé et al., who found that flow cytometry analysis confirmed a senescence phenomenon with an S phase cycle arrest after two days of treatment at the corresponding P15 IC_50_, 1 µM (Figure 6b) [55].

Moreover, flow cytometry analysis allowed determination of the effect of concentration on the apoptotic or senescence pathways. Indeed, Lainé et al. showed that at 0.1 µM, DP1-loaded LNCs induced cell cycle blockage in the S phase in 9L glioma cells, a characteristic of senescence, whereas at a higher concentration (0.5 µM), DP1-loaded LNCs blocked cells in the G0/G1 phase and led to apoptosis. This was confirmed by the counting of the apoptotic cells: 15% of apoptotic cells at 0.5 µM of DP1 compared to 5% at 0.1 µM [55]. Using a different method, Topin-Ruiz et al. demonstrated that P722-loaded LNCs potentiated apoptosis via the intrinsic pathway: increase in procaspase 9, cleaved caspase 9 and procaspase 3 in B16F10 melanoma cells after treatment, opening up a new therapeutic strategy with encapsulated P722 [96].

Finally, while ferrocifens demonstrated outstanding effects on various cancer cell lines, it is important to note that these compounds resulted in little or no toxicity on healthy cells, such as melanocytes and astrocytes, even after encapsulation into LNCs [37,39]. Indeed, Allard et al. showed that at a concentration up to 10 µM for P5, the cell viability of astrocytes was still higher than 80%, whereas the IC_50_ of P5 on glioma 9L cells was about 0.5 µM. This phenomenon was explained by the fact that P5 could take advantage of the microenvironment of the cancerous cell, rich in ROS and leading to the corresponding QM, whereas the high amount of antioxidant proteins such as glutathione in astrocytes allowed an efficient scavenging of ROS. Another explanation advanced by the authors would be due to the easier accessibility of ferrocifens to the intracellular target during cell division. The same result was observed with conventional loaded LNCs, confirming the theory that the low toxicity observed in healthy cells is due to the mechanism of action of these organometallic compounds and not to an active targeting strategy. However, in some studies, the surface modification of the LNCs with active ligands (antibody, peptide, etc.) was studied, either because they did not contain ferrocifens [26] or because the recognition of certain cancer cells by active targeted LNCs was expected to allow an even more specific accumulation in the tumor cells, as observed, for example, with NFL P53 LNCs [95].

## 5. Ferrocifen-Loaded Lipid Nanocapsules: In Vivo Experiments

All the in vivo studies carried out on ferrocifen-loaded lipid nanocapsules (LNCs) are summarized in Table 4 below. Several ferrocifens (P5, P15, DP1, P53 and P722) were successfully encapsulated in LNCs. The efficiency of these drug delivery systems on several cancer cells lines (glioma, melanoma, breast cancer) was discussed. No cytotoxicity or hepatotoxicity was found, whatever the number of injections, administration route or formulation strategy. 

### 5.1. Glioblastoma

Malignant brain tumors are among the most severe cancers due to their poor diagnosis and life expectancy. Glioblastoma is the most common and aggressive type of primary malignant brain tumor, resulting in the assignment of grade IV, the highest grade in the World Health Organization classification [100]. Glioblastoma remains a difficult cancer to treat; surgery followed by radiotherapy and/or chemotherapy only slightly increases survival rates. Indeed, even if some improvements in diagnosis and treatments have been made, the survival rate still remains low, with most patients dying within two years after diagnosis. One of the main problems in treating glioblastoma is the need to cross the blood–brain barrier (BBB). Thus, discovering new strategies and treatments to cross the BBB and target the tumoral microenvironment remains a challenge. For example, in recent decades, convection-enhanced delivery (CED) has been developed (in the early 1990s) and is now considered as a promising administrative route [101,102]. This technique is a local delivery method (direct intracranial delivery) that enhances the drug brain distribution by continuously injecting a therapeutic fluid under positive pressure [102,103]. According to MacKay et al. [103], the ideal nanoparticle for CED would be smaller than 100 nm in diameter with a neutral or negative surface charge. Thus, LNCs seem to have the required characteristics as they can be formulated with a diameter of around 50 nm, with a zeta potential that is often negative. Allard et al. and Huynh et al. demonstrated that using P5-loaded LNCs for CED injection was promising but strongly dependent on the formulation strategy [39,97]. Indeed, several studies demonstrated the importance of the formulation process chosen (conventional P5-LNCs; stealth P5-LNCs; peptide-coated P5-LNCs) depending on the administration route: local administration via CED, intravenous injection or intracarotid injection. Indeed, Huynh et al. obtained an increase in the median survival time (28 days compared to 25 days for untreated group) of orthotopic 9L glioma-bearing rats after CED injection of conventional P5-LNCs compared to stealth P5-LNCs. The opposite was observed after a single intracarotid injection (30 days with the stealth P5-LNCs compared to 27 days with the conventional LNCs) [97], as shown in Figure 7a. These results can be linked to the PEGylation paradox. On the one hand, PEG allows an increase in blood circulation time, avoiding the opsonization phenomenon and resulting in a higher accumulation of nanocarriers at the tumor site. On the other hand, PEG can prevent the cell internalization of nanocarriers [16]. The passive targeting strategy is thus more suitable for a systemic injection than for a local injection. In another study, Lainé et al. made the same observation: the administration route had to be decided based on the formulation strategy. The authors formulated NFL peptide-coated P5-loaded LNCs and injected them into an orthotopic rat model (9L glioma cells) by a single intracarotid or by CED injections. No improvement in the median survival rate was obtained by CED injection, compared to the intracarotid injection, for which the survival of rats was improved, with some animals surviving until 44 days with a treatment by NFL peptide-coated P5-LNCs (a median survival time of 27 days for the untreated rats was obtained) [28]. It is a clear improvement compared to the 30 days of median survival time obtained with stealth P5-LNCs and the 27 days obtained for the conventional P5-LNCs in the same orthotopic model [97]. These results, shown as Kaplan–Meier plots in Figure 7a, demonstrated the advantage of active targeting for systemic administration, by using peptide-coated ferrocifen LNCs.

Finally, it is well-known that repeated intravenous injections may allow the increase in the drug concentration in tumor tissue but may lead to some side effects such as the Accelerated Blood Clearance (ABC) phenomenon and hepatotoxicity [104]. Even if only one study was performed on this ABC phenomenon with ferrocifens, the results obtained by Lainé et al. were unusual. Indeed, the second injection of stealth DP1-loaded LNCs did not show any drastic decrease in LNC blood circulation time, as shown in Figure 7b, in comparison with several studies where PEGylation of nanocarriers such as liposomes or gold NPs induced an immune response, leading to a rapid clearance of the nanocarriers [99,104,105]. Moreover, repeated intravenous injections (10 injections) of stealth DP1-LNCs did not induce any hepatotoxicity according to immunohistological analyses. A promising tumor volume reduction was even obtained, from 1400 mm^3^ for untreated rats to 300 mm^3^ for treated rats after 21 days (Figure 7c), showing that this drug delivery nanocarrier could be injected into animals several times with good benefits. These results could be explained by the senescence phenomenon, with an S phase arrest evidenced during the cell cycle and confirmed by previous observations made with DP1, such as the inhibition of pluripotent cancer cells [106]. Thus, all the studies performed on ferrocifen-loaded LNCs showed very promising results in glioma models.

### 5.2. Other Cancers

#### 5.2.1. Breast Cancer

Breast cancer is one of the most common cancers, corresponding to 15% of all cancer deaths among women. Metastasis to other organs such as the liver, lung, brain and bones, is frequent and accounts for the majority of deaths from this cancer [107,108]. Hopefully, the mechanism involved in breast cancer metastasis is now better understood (discovery of genes related to breast cancer), aiding the development of new therapeutic strategies. The estrogen receptor (ER) is known as a major target for breast cancer chemotherapy. Indeed, more than 70% of breast cancers are ERα+. This is the reason selective estrogen receptor modulators (SERMs) have been developed in recent decades. The best known SERM is tamoxifen (TAM). However, one major drawback of this chemotherapeutic molecule is its selective activity against only hormone-dependent estrogen receptor ERα+ cases and not hormone-independent estrogen receptor ERα- cases. Moreover, TAM therapy presents some side effects such as risk of endometrial cancer, pulmonary embolism, stroke and deep-vein thrombosis [109]. As previously mentioned, ferrocifens were effective in vitro against breast cancer cell lines such as MCF-7 cells (ERα+ cells) and MDA-MB-231 cells (ERα- cells). The in vivo efficacy of P15 was explored in triple-negative breast cancer (TNBC) cell lines, MDA-MB-231 cells, using stealth P15-LNCs [55]. An ectopic model was chosen and two intraperitoneal injections of the suspension were applied. This study provided evidence that blank stealth LNCs did not have any biological effect on the MDA-MB-231 cells compared to the stealth P15-LNCs, confirming the in vitro results obtained on the nontoxicity of LNCs. A significant 36% decrease in the tumor volume was found for mice treated with stealth P15-LNCs after 24 days of treatment injection compared to the blank stealth LNC control (Figure 8a). This tumor volume reduction was associated with the cytostatic effect of the stealth P5-LNCs on MDA-MB-231 cells, as shown in vitro by flow cytometry, with an S cell cycle arrest. This is the first result evidenced in vivo with ferrocifens on this highly resistant cancer and it appears to be very promising [55].

#### 5.2.2. Metastatic Melanoma

Malignant melanoma is the leading cause of death from skin cancer with a 5-year survival rate of less than 10%. Normally, surgery is an adequate treatment when the tumor is diagnosed early enough. A chemotherapy treatment with dacarbazine (DTIC) is additionally applied, with a low response rate [110]. There is a real need to overcome the high resistance phenomena of melanoma cells against chemotherapy. Thus, Resnier et al. decided to test ferrocifen-LNCs against this pathology. They showed promising results with ferrocifen-loaded LNCs on metastatic melanoma compared to dacarbazine treatment. Stealth DP1-loaded LNCs and stealth P5-loaded LNCs were intravenously injected (1 injection per day, 5 days consecutively) for 3 weeks on an ectopic murine model of human SK-Mel28 melanoma cells. The results demonstrated that stealth DP1-loaded LNC treatment was more efficient than stealth P5-loaded LNCs and DTIC treatment. Indeed, the tumor volume reduction was better with the use of formulated DP1, 21 days after the first injection, as shown in Figure 8b. The tumor volume was reduced by 40% with stealth DP1-LNCs and 9% with stealth P5-LNCs compared to the untreated group and a difference of around 13% was observed between stealth DP1-LNCs and DTIC treatments. This result confirmed the highest sensitivity of DP1 observed in vitro (compared to P5) on human SK-Mel28 melanoma cells (IC_50_ = 1 µM for DP1; IC_50_ = 3 µM for P5 and IC_50_ > 100 µM for DITC) [59].

Moreover, as previously mentioned for 9L glioma cells [99], this study confirmed that LNCs did not present any hepatoxicity. Indeed, after several injections, the amount of aspartate aminotransferase (ASAT) and alanine aminotransferase (ALAT), enzymatic biomarkers of liver inflammation, were similar between blood samples from untreated mice and blood samples from treated mice [59].

Finally, it was shown for the first time, in another recent study, that some ferrocifens could induce a new therapeutic pathway in resistant melanoma cancer. Indeed, as previously described, P722-loaded LNCs were used on an orthotopic murine model of human B16F10 melanoma. In addition to an induction of apoptosis via the intrinsic pathway, the results revealed a significant increase in activated CD_8_^+^ T lymphocytes after treatment of stealth P722-LNCs, leading to immunogenic cell death [96].

## 6. Association with Other Therapies: An Interesting Perspective

### 6.1. Ferrocifens and Radiotherapy

As previously described, glioblastoma remains difficult to treat despite improvements in chemotherapy and radiotherapy [111,112,113]. In previous work, Allard et al. showed that using LNCs to encapsulate a lipophilic radiopharmaceutical compound for internal radiotherapy could be promising [32]. External sources of radiation remain the chief radiotherapeutic treatment method; however, ferrocifen-loaded LNCs were indeed used in combination with external radiotherapy on an orthotopic 9L glioma model in rats. After administration of P5-loaded LNCs by CED at a corresponding dose of 0.36 mg/rat, followed by three irradiations of 6 Gy doses, a synergic effect between ferrocifen treatment and radiotherapy was observed. With only chemotherapy treatment using P5-loaded LNCs, the median survival time of rats was improved: 25 days for the untreated rats and 27.5 days for the treated rats [58,88]. Better yet, combination with external irradiation treatments led to a clear improvement: the median survival time of rats was 40 days, similar to that for NFL-coated P5-LNCs [28,58]. Local irradiation was able to potentiate the cytotoxic action of P5 with the production of ROS inside the tumor microenvironment, allowing the formation of the corresponding QM. Indeed, it is well-known that the level of ROS in the brain is low due to the high concentration of antioxidants. Thus, ionizing radiation probably allowed the formation of ROS followed by the oxidation of Fe(II) to Fe(III) of the ferrocifen. Ferrocifen-loaded LNCs could thus be used as an efficient radiosensitive anticancer drug delivery system.

### 6.2. Ferrocifens and Gene Therapy

Several studies highlighted the idea that overexpression of Bcl-2 could be responsible for the chemoresistance of melanoma [114,115]. Bcl-2 is a gene regulating the apoptotic pathway by blocking the oligomerization of proapoptotic proteins such as Bax and Bak. Small interfering RNA (siRNA)-mediated gene therapy could be a promising strategy to inhibit the expression of Bcl-2 in melanoma. However, one major problem of siRNA is its low bioavailability in the bloodstream and incapacity to diffuse through the cell membrane [116]. Nevertheless, it has been demonstrated that LNCs could be used as DNA and siRNA delivery systems [31,89,117,118,119]. Previously, Resnier et al. showed that siRNA preformulated in lipoplexes could be incorporated in stealth LNCs with an encapsulation efficiency of around 60%, and no cytotoxicity of this nanocarrier was seen in SK-Mel28 melanoma cells [85]. The same authors then decided to combine gene therapy and chemotherapy by the association of Bcl-2 siRNA and stealth DP1-loaded LNCs to treat SK-Mel28 melanoma cells. DP1 was used for the in vivo study as it was the most efficacious in SK-Mel28 melanoma cells compared to P5, as mentioned previously [59,85]. After repeated intravenous injections of the stealth siBcl-2/DP1-LNCs for one week followed by two weeks of stealth DP1-LNC treatment, a synergic effect was observed in an ectopic murine model. This allowed a tumor reduction of about 50% compared to 25% for an individual administration of siBcl-2-LNCs and stealth DP1-LNCs. Bcl-2 siRNA would inhibit Bcl-2 expression, allowing melanoma cells to respond to the action of DP1, thereafter inducing apoptosis as well as senescence [59]. This was the first confirmation that ferrocifen-loaded LNCs could be very promising nanocarriers combined with gene therapy (Figure 8b)) in a multitherapy strategy.

### 6.3. Ferrocifen-LNC-Loaded Cellular Vectors

Another potent strategy to target cancer cells with NPs is to use mesenchymal stromal cells (MSCs), such as marrow-isolated adult multilineage inducible (MIAMI) cells, a subpopulation of MSCs. In glioblastoma studies, it has been highlighted that MIAMI cells were mainly localized at the border between tumor cells and healthy brain cells. For this application, LNCs were efficiently internalized into MIAMI cells without any negative effects on cell viability and differentiation, allowing their migration towards U87MG glioma cells. However, such modified MIAMI cells did not modify the survival of U87MG-bearing mice [98,120,121,122]. In further studies, ferrocifens were encapsulated into LNC-loaded MIAMI cells in order to target U87MG glioma cells [57,98]. The internalization of the P5-loaded LNCs in MIAMI cells did not induce any cytotoxic effect on MIAMI cells. A single intratumoral injection of this new vector on an orthotopic U87MG glioma model in nude mice resulted in a significant reduction in the tumor volume [98]. This was confirmed by a study of Clavreul et al., where they compared the efficacy of a single intratumoral injection of P5 LNC-loaded MIAMI cells (treatment 1) with the efficiency of a single intratumoral injection of P5-loaded LNCs (treatment 2). Magnetic resonance imaging (MRI) indicated that treatment 1 induced a better reduction in the tumor volume. Even though the median survival time was still modest (29 days), this increase was obtained with a single injection (dose of 3.6 µg/animal) [57], whereas chemotherapy with ferrocifen-loaded LNCs usually involved several injections [99], or a higher dose of treatment (2.4 mg/animal with a CED injection, 0.36 mg/animal with an intravenous injection) [28,39,88]. Combining stem cell therapy with ferrocifen-loaded LNCs could be another promising alternative tool.

## 7. Conclusions

Since their first synthesis at the end of the 1990s, ferrocifens, molecules of the organometallic family of compounds, have been under constant investigation to understand their mechanisms of action. They have shown remarkable in vitro antiproliferative and cytotoxic effects on various cancer cell lines. Their mechanisms of action are different from those of the usual chemotherapeutic compounds (cisplatin, doxorubicin, paclitaxel and tamoxifen), well-known DNA-damaging or microtubule-stabilizing agents and SERMs. Indeed, ferrocifens are molecules endowed with multiple biological properties, including anticancer activity, that induce inhibition of cancer cell growth through apoptosis and/or senescence according to the conditions previously described (structure of the complexes, ferrocifen concentration, cell lines tested depending on whether they are healthy or cancerous). Additionally, by taking advantage of the microenvironment of the tumor (acid pH and oxidative stress), these organometallic compounds might overcome the resistance of MDR cancer to apoptosis. Moreover, ferrocenyl quinone methide, which has selective electrophilic reactivity, can interact with thioredoxin reductase and so inhibit its activity, leading to the production of ROS. These promising in vitro results have been confirmed widely in in vivo experiments by using LNCs as nanocarriers to encapsulate the highly lipophilic ferrocifens. Whether on ectopic or orthotopic models, whatever the cancer cell lines tested—glioblastoma, breast cancer, metastatic melanoma, all well-known for their chemoresistance to current therapy—in vivo treatment by ferrocifen LNCs revealed, in most cases, an increased median survival rate in animal models. The rationale for combining therapies using drugs or other strategies (gene therapy, radiotherapy, stem cell therapy, etc.) operating via different mechanisms and/or acting synergistically, and thus decreasing the likelihood that resistant cancer cells will develop, was also successfully tested with ferrocifen LNCs.

However, even if there is evidence for the effictiveness of these nanomedicines, it is important to keep a critical eye on the results obtained as they can vary depending on the selected model—ectopic or orthotopic. For example, it is well-known that orthotopic tumor models develop a tumor microenvironment closer to the reality than the ectopic tumor models [123]. Moreover, for most of the studies, the cell lines were not directly derived from patients. Thus, to go deeper into the promising preclinical studies with ferrocifen-loaded LNCs, it should be interesting to perform experiments in vivo on PDX models to reduce, as far as possible, the gap between human tumors and preclinical models, so as to be more representative of the tumor microenvironment of human patients [124,125].

## Figures and Tables

**Figure 1 cancers-13-02291-f001:**
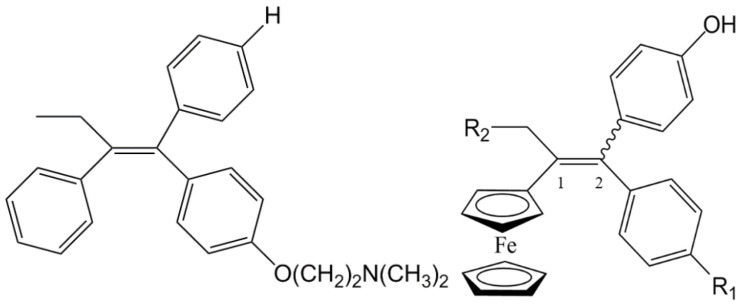
General structure of tamoxifen and ferrocifen compounds.

**Figure 2 cancers-13-02291-f002:**
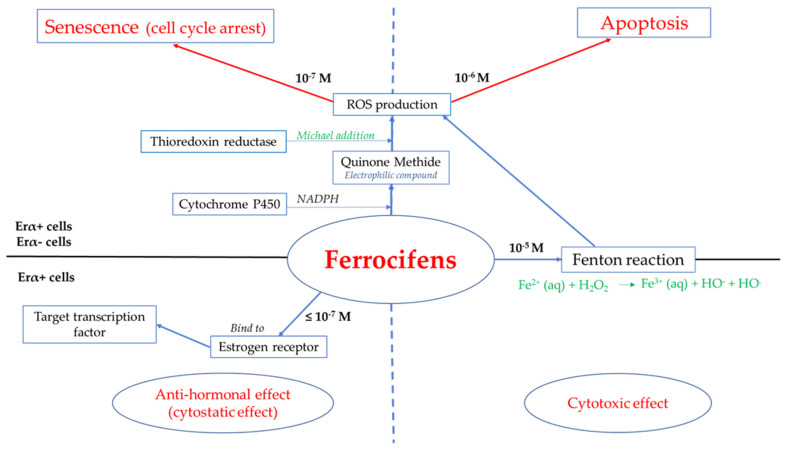
Ferrocifen mechanisms of action at the cellular level: at a concentration ≤ 10^−7^ M, ferrocifens present a cytostatic effect; at a concentration > 10^−7^ M, ferrocifens present a cytotoxic effect.

**Figure 3 cancers-13-02291-f003:**
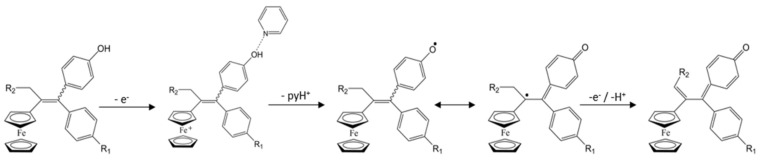
Mechanism of quinone methide formation from ferrocifen. This mechanism was confirmed by electrochemistry and electron paramagnetic resonance.

**Figure 4 cancers-13-02291-f004:**
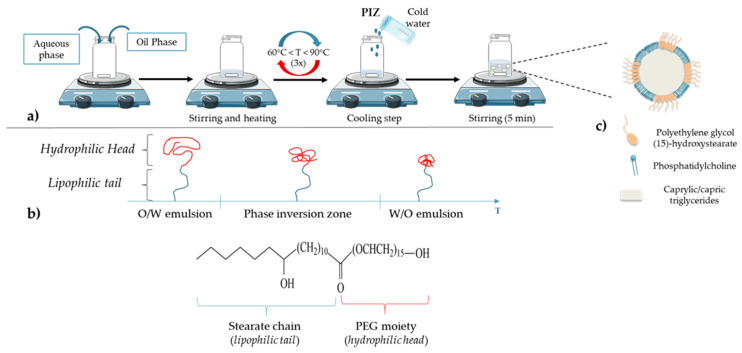
(**a**) Lipid nanocapsule (LNC) formulation: a free organic solvent process; (**b**) schematic behavior of polyethylene glycol (15)-hydroxystearate under temperature variation (polyethylene glycol (15)-hydroxystearate becomes more hydrophobic when the temperature increases, which leads to the phase inversion from oil in water to water in oil emulsions); (**c**) general structure of LNC: an oily core composed of caprylic/capric triglycerides surrounded by a shell made of phosphatidylcholine from soybean and polyethylene glycol (15)-hydroxystearate.

**Figure 5 cancers-13-02291-f005:**
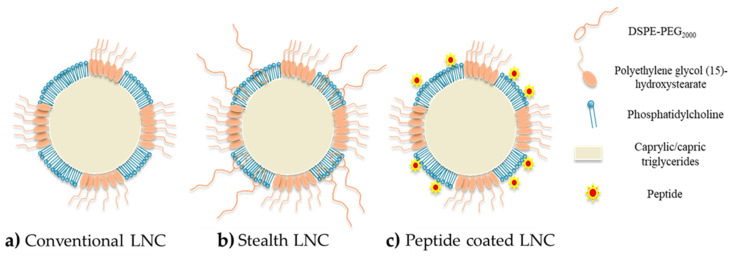
Schematic structure of LNCs: (**a**) conventional LNCs (without surface modification); (**b**) stealth LNCs with surface modification by DSPE-PEG_2000_; (**c**) active targeting LNCs with surface modification by peptides.

**Figure 6 cancers-13-02291-f006:**
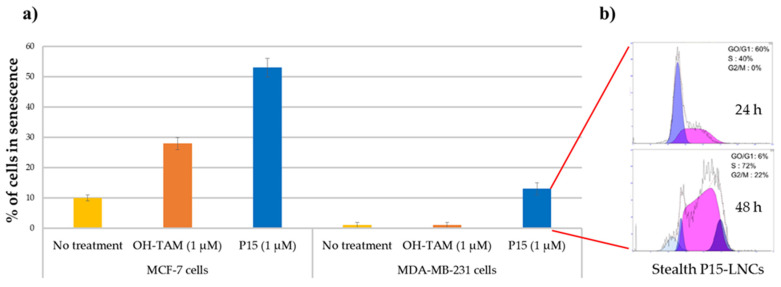
(**a**) Percentage of cells in senescence obtained by SA-galactosidase staining, 5 days after treatment on MCF-7 cells and MDA-MB-231 cells (no treatment as control in yellow; OH-TAM at 1 µM in orange; P15 at 1 µM in blue); (**b**) flow cytometric analysis profile of the MDA-MB-231 cell cycle 24 and 48 h after stealth P15-LNC treatment. A cell cycle arrest in S phase was observed corresponding to a senescence phenomenon (adapted with permission from [54,55], Copytight 2021 Elsevier).

**Figure 7 cancers-13-02291-f007:**
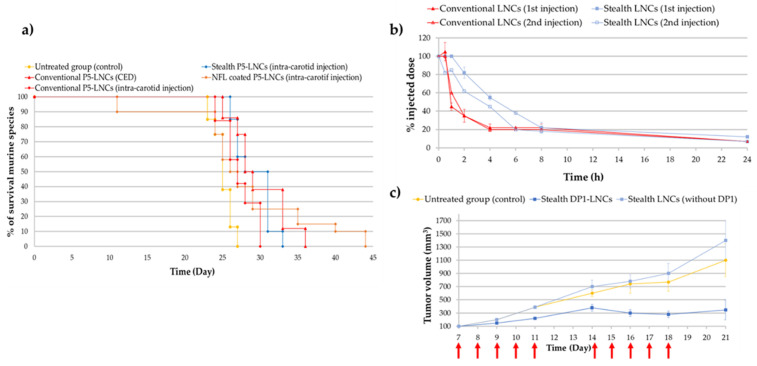
(**a**) Kaplan–Meier survival plots for 9L glioma-bearing rats receiving CED injection of conventional P5-LNCs (red triangle) or intracarotid injection of NFL-coated P5-LNCs (orange diamond), stealth P5-LNCs (blue diamond) and conventional P5-LNCs (red diamond). The red arrow indicates the injection day: day 6 (adapted with permission from [28,97], Copytight 2021 Elsevier); (**b**) ABC phenomenon evaluation of stealth LNCs between the first (blue square) and second injection (unfilled blue square) (adapted with permission from [99], Copytight 2021 Elsevier); (**c**) in vivo treatment efficiency of stealth DP1-LNCs (dark blue square) on ectopic 9L glioma-bearing rats, the red arrows indicated the days of injections (adapted with permission from [99], Copytight 2021 Elsevier).

**Figure 8 cancers-13-02291-f008:**
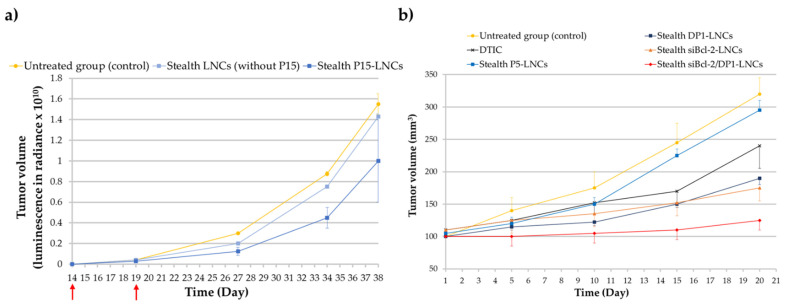
(**a**) Tumor volume evolution after intraperitoneal injections of stealth P15-LNCs in mice bearing ectopic MDA-MB-231 cells. The red arrows indicate the days of treatment. Bioluminescence has been measured in radiance (pixels/second/cm^2^/square) in order to obtain the tumor volume (adapted with permission from [55], Copytight 2021 Elsevier); (**b**) tumor volume evolution after repeated intravenous injections of stealth DP1-LNCs (blue dark square), P5-LNCs (blue square), dacarbazine/DTIC (dark cross), stealth siBcl-2-LNCs (orange triangle) and stealth siBcl-2/DP1-LNCs (red diamond) on SK-Mel28 cells bearing mice. The abscissa axis starts after the treatment injections (adapted with permission from [59], Copytight 2021 Elsevier).

**Table 1 cancers-13-02291-t001:** Structure of P5, P15, P53, DP1 and P722 ferrocifens and their corresponding quinone methides.

Ferrocifen	Quinone Methide
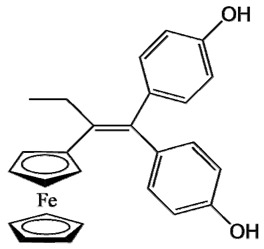 **P5 (FcdiOH)**	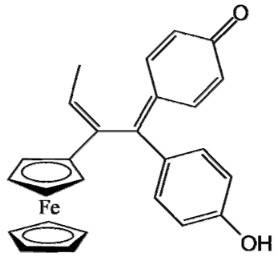 **Vinyl-QM-P5**
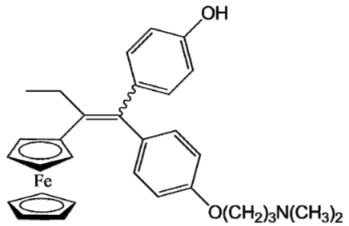 **P15 (FcOHTam)**	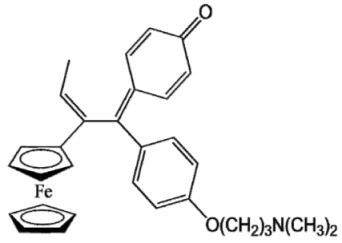 **Vinyl-QM-P15**
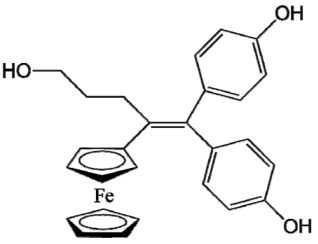 **P53 (FcTriOH)**	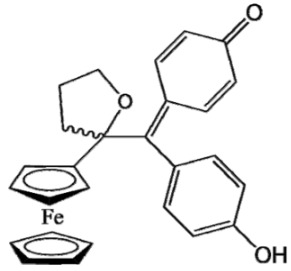 **Furane-QM-P53**
	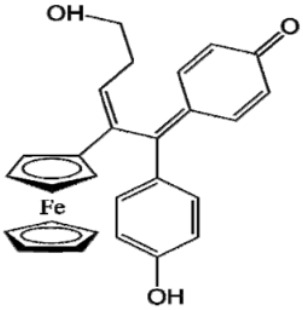 **Vinyl-QM-P53**
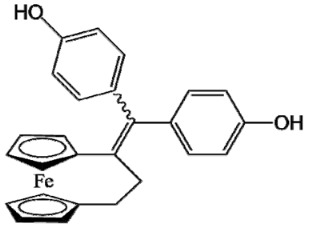 **DP1 (Ansa-FcdiOH)**	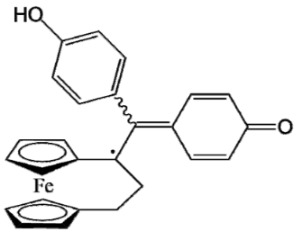 **Radical DP1**
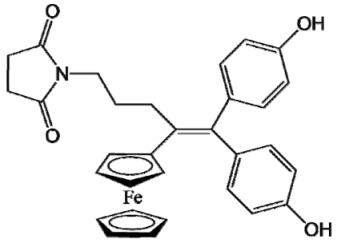 **P722**	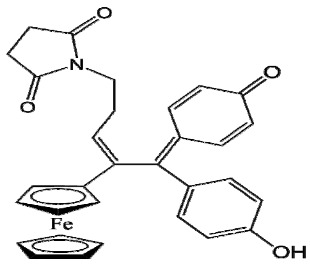 **QM-P722**

**Table 2 cancers-13-02291-t002:** IC_50_ values in µM on some cancer cell lines (breast, glioblastoma, melanoma and ovarian) of P5 and P15 ferrocifens.

	Cell Line	P5	P15
Breast	MCF7	[55]	[55]
	MDA-MB-231	[56]	[56]
Glioblastoma	U87MG	[57]	-
	9L	[58]	-
Melanoma	WM35	[37]	[37]
	WM793	[37]	[37]
	WM9	[37]	[37]
	SK-Mel28	[59]	-
Ovarian	A2780	[56]	-
	A2780cisR	[56]	-
	IC_50_ (µM)		
	0.5–2		
	2–10		

**Table 3 cancers-13-02291-t003:** GI_50_ values in µM on some cancer cell lines (breast, central nervous system (CNS) cancer, melanoma, ovarian cancer and leukemia) of the DP1, P53 and P722 ferrocifens.

	Cell Line	DP1 [51]	P53 [50]	P722 [48]
Breast	MCF7			
	MDA-MB-231			
CNS	SF-295			
	SF-539			
	SF-268			
Melanoma	SK-MEL-2			
	SK-MEL-28			
	SK-MEL-5			
Ovarian	IGROV1			
	SK-OV-3			
Leukemia	HL-60(TB)			
	MOLT-4			
	GI_50_ (µM)			
	<0.1			
	0.1–0.5			
	0.5–2			
	2–10			
	>10			

**Table 4 cancers-13-02291-t004:** In vivo studies performed with P5-, P15-, P53- and DP1-loaded LNCs on glioblastoma, metastatic melanoma and breast cancers.

Ferrocifens Used (Names)	Pathology	LNC Formulation	Encapsulation Efficiency (Drug Loading)	In Vivo Studies
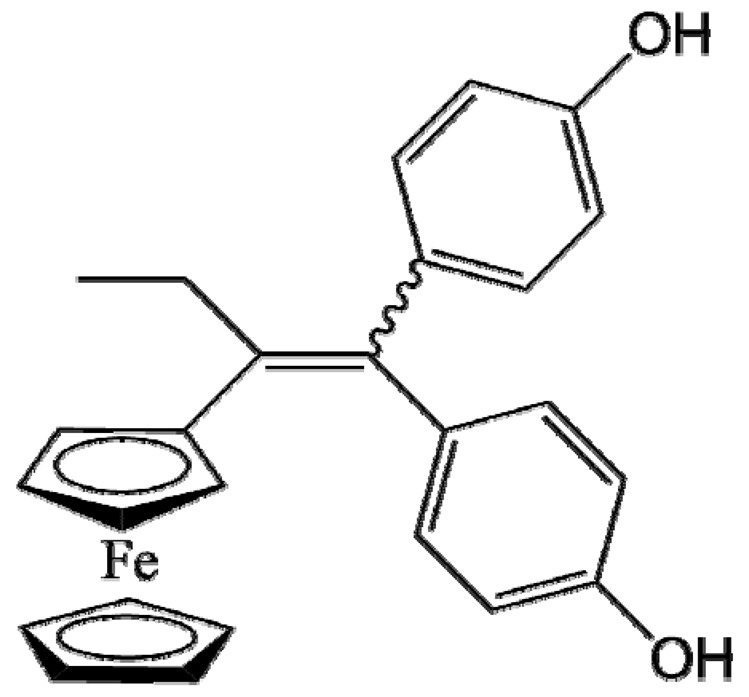 **P5 (FcdiOH)**	Glioblastoma (9L glioma cells)	Conventional LNCs	>98% (1.0 mg/g or 6.5 mg/g)	Single intratumoral injection (400 µL); ectopic model [39]
	Glioblastoma (9L glioma cells)	Conventional LNCs	96% (1.0 mg/g or 6.5 mg/g)	Single intracranial injection by CED (60 µL, 0.36 mg/rat) + 3 irradiations of 6 Gy doses; orthotopic model [58]
	Glioblastoma (9L glioma cells)	NFL-TBS-63 peptide-coated LNCs	(6.1 mg/mL)	Intracarotid injection (400 µL, 2.5 mg/rat); orthotopic model [28]
			(5.5 mg/mL)	Injection by CED (60 µL, 0.36 mg/kg rat); orthotopic model [28]
	Glioblastoma (9L glioma cells)	Stealth LNCs	>98% (6.5 mg/g)	Single intravenous injection (400 µL, 2.4 mg/rat); ectopic model [88]
				Single intravenous injection (400 µL, 2.4 mg/rat); orthotopic model [88]
	Glioblastoma (9L glioma cells)	Stealth LNCs and conventional LNCs	(6.5 mg/g)	Intracarotid injection (400 µL, 2.4 mg/rat), orthotopic model [97]
				CED injection (60 µL, 0.36 mg/rat), orthotopic model [97]
	Glioblastoma (U87MG cells)	LNC-loaded MIAMI cells	(2.6 mg/mL)	Intratumoral injection (100 µL), heterotopic model [98]
	Glioblastoma (U87MG cells)	LNC-loaded MIAMI cells	(6.0 mg/g of LNCs) 20 pg of P5/MIAMI cells	Intratumoral injection (3.6 µg/mouse), orthotopic model [57]
	Melanoma (SK-Mel28 human cells)	Stealth LNCs	92% (6.0 mg/mL)	Repeated intravenous injection (45 mg/kg); ectopic model [59]
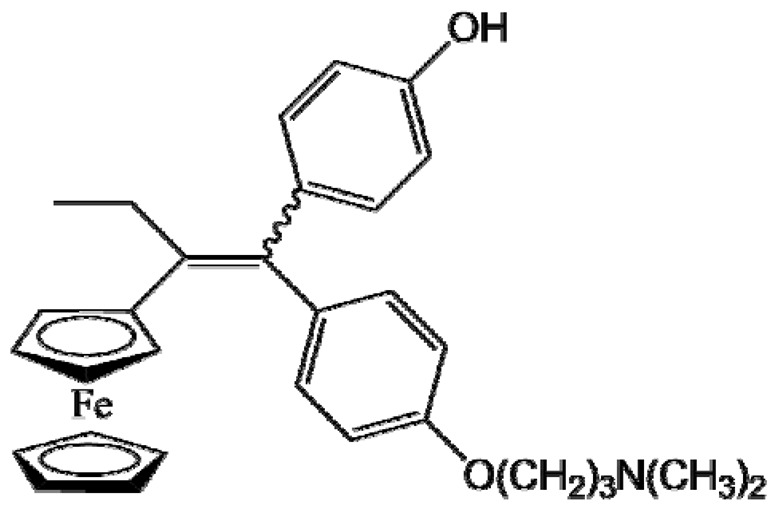 **P15 (FcOHTam)**	Breast cancer (MDA-MB-231 cells)	Stealth LNCs	100% (8.0 mg/mL)	Repeated intraperitoneal injection (2×) (20 mg/kg); ectopic model [55]
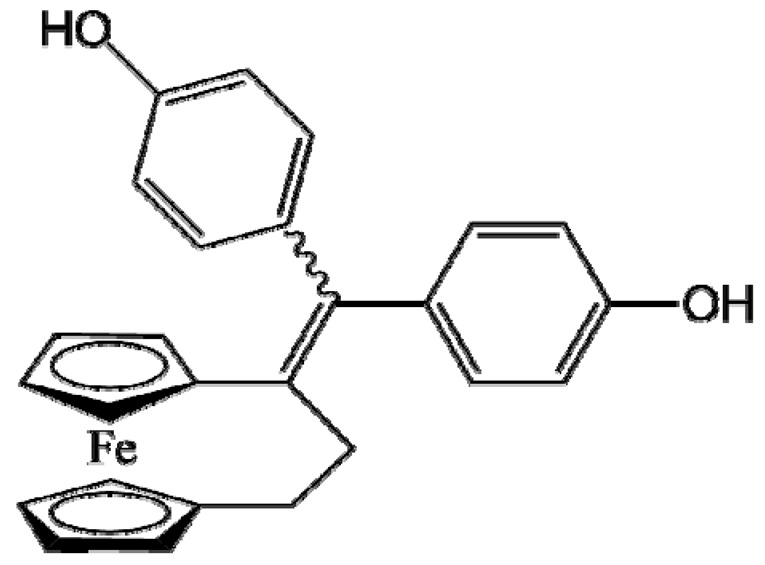 **DP1 (Ansa-FcdiOH)**	Melanoma (SK-Mel28 human cells)	Stealth LNCs	96% (6.0 mg/mL)	Repeated intravenous injection (45 mg/kg); ectopic model [59]
		Stealth LNCs with Bcl-2 siRNA	85% (6.0 mg/mL)	Repeated intravenous injection (45 mg/kg); ectopic model [59]
	Glioblastoma (9L glioma cells)	Stealth LNCs	(6.4 mg/g)	Repeated intravenous injection (10×) (20 mg/kg); ectopic model [99]
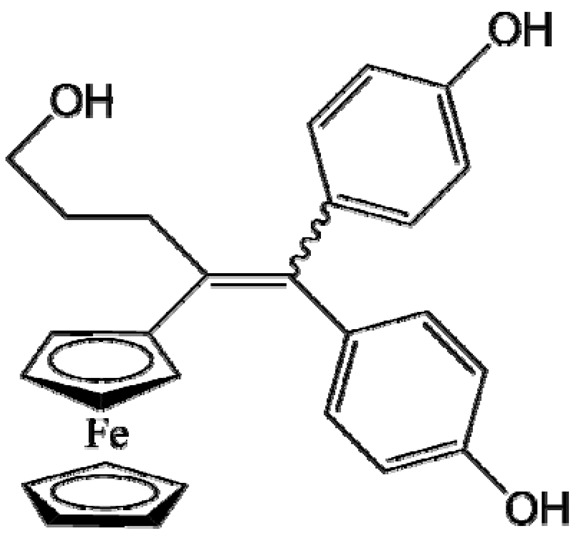 **P53 (FctriOH)**	Glioblastoma (U87MG cells)	NFL-TBS40-63 peptide-coated LNCs	>99% (2.67 mg/g)	Intravenous injection (2×) (70 µL, 20 mg/kg); ectopic model [95]
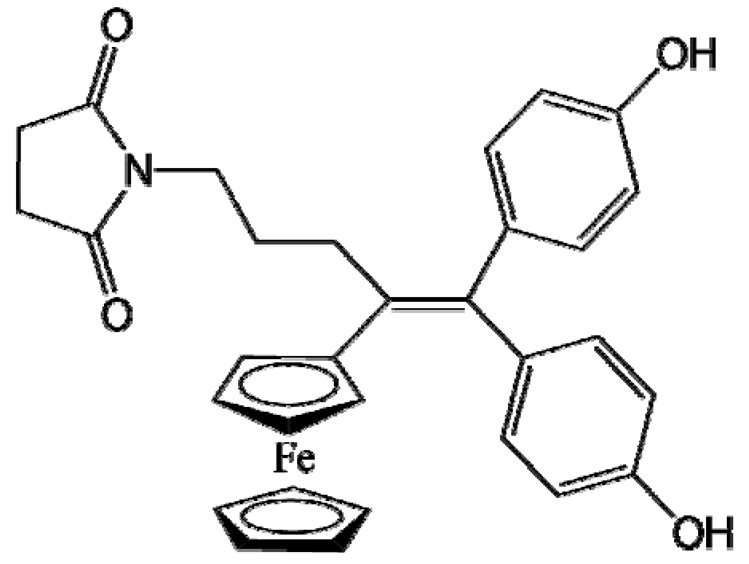 **P722**	Melanoma (B16F10 cells)	Stealth LNCs	65%	Intraperitoneal injection (7 mg/kg); orthotopic model [96]

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
