# Peer review of "Ferrocifen Loaded Lipid Nanocapsules: A Promising Anticancer Medication against Multidrug Resistant Tumors"

_cancers, 2021, doi:10.3390/cancers13102291_

Round 1
Reviewer 1 Report
The review manuscript number 1200616, entitled “Ferrocifen loaded lipid nanocapsules: a promising anticancer medication against multidrug resistant tumors”, presents an overview of the in vitro and in vivo studies performed with ferrocifen loaded in lipid nanocapsules on several multidrug resistance cancers. The manuscript focus a relevant thematic, it is well organized, systematic and objective. But some points should be considered and improved before publication.
Minor points:
- In my point of view, the authors should also assume in the Simple Summary that the manuscript will give an overview of in vitro and in vivo studies performed with ferrocifen…
- Please improve beginning and clarify the sentence of line 26 in the Abstract.
- According your manuscript organization and the content of sentence line 29 in the Abstract, probably the authors should refer in vitro experiments … in cancer cell lines…!
- Figure 4 c) correct triglycerides writing.
- Improve the sentence in line 322 avoiding the term “thanks”.
- Improve the formatting of some figures and Tables, for instance figure 6, using the same font.
- The authors use too old references, when possible, they should be updated for new/more recent ones.
Major points:
- The authors refer in topic 3.2 some surface modifications for passive or active targeting. But then, in the end of topic 4, it is not clear why the ferrocifens demonstrated outstanding effects on various cancer cell lines but did not lead any toxicity on healthy cells. Can be due to the mechanism of action that only occurs in cancer cells? Or the systems are functionalized to only recognize the cancer cells? Please. Concretize the ideas. There are some studies performed with the aim of modifying the LNCs surface to only target/transfect the cancer cell in a specific cancer? If yes, please include or clarify in the manuscript.
- In Conclusion section, the authors refer the mechanisms of action of ferrocifen compounds are different form the usual chemotherapeutic compounds. But, it is not clear what are the differences. What are the mechanisms of action of these usual compounds? Why they are not efficient in multidrug resistance cancers? Probably including a briefly information in the introduction section, when these usual compounds are mentioned by the first time, or in another appropriate place of the manuscript, could help to better understand the advantages and specificities that bring the ferrocifen compounds to overcome the multidrug resistances.
Reviewer 2 Report
This review manuscript is fabulous, and it is obvious that the authors have spend a great deal of time putting it together with great detail and graphical representations. I only have a few minor corrections to suggest:
1)The English language throughout the manuscript needs improving, but not by much.
2) I noticed a lot of the figures are adapted from different references and SMART artwork is also used- I would ensure hat the correct permissions have been conformed for the use of these materials.
Excellent work.
